# High-performance and compact-designed flexible thermoelectric modules enabled by a reticulate carbon nanotube architecture

Wenbin Zhou[1,2,3], Qingxia Fan[1,3], Qiang Zhang[1,3], Le Cai[1,3,†], Kewei Li[1,3], Xiaogang Gu[1,3,4], Feng Yang[1,3,4], Nan Zhang[1,3], Yanchun Wang[1,3,4], Huaping Liu[1,3,4], Weiya Zhou[1,3,4] & Sishen Xie[1,3,4]

It is a great challenge to substantially improve the practical performance of flexible thermoelectric modules due to the absence of air-stable n-type thermoelectric materials with high-power factor. Here an excellent flexible n-type thermoelectric film is developed, which can be conveniently and rapidly prepared based on the as-grown carbon nanotube continuous networks with high conductivity. The optimum n-type film exhibits ultrahigh power factor of $\sim 1,500\,\mu W\,m^{-1}\,K^{-2}$ and outstanding stability in air without encapsulation. Inspired by the findings, we design and successfully fabricate the compact-configuration flexible TE modules, which own great advantages compared with the conventional $\pi$-type configuration modules and well integrate the superior thermoelectric properties of p-type and n-type carbon nanotube films resulting in a markedly high performance. Moreover, the research results are highly scalable and also open opportunities for the large-scale production of flexible thermoelectric modules.

[1] Beijing National Laboratory for Condensed Matter Physics, Institute of Physics, Chinese Academy of Sciences, Beijing 100190, China. [2] Institute of Engineering Thermophysics, Chinese Academy of Sciences, Beijing 100190, China. [3] Beijing Key Laboratory for Advanced Functional Materials and Structure Research, Beijing 100190, China. [4] University of Chinese Academy of Sciences, Beijing 100049, China. † Present address: Department of Electrical and Computer Engineering, Michigan State University, East Lansing, Michigan 48824, USA. Correspondence and requests for materials should be addressed to W.Z. (email: wyzhou@iphy.ac.cn) or to S.X. (email: ssxie@iphy.ac.cn).

Thermoelectric (TE) modules composed of several p–n couples can directly convert thermal energy generated from natural heat sources or ubiquitous waste heat to valuable electric energy[1–3]. To make great contributions to global sustainable energy solutions, increasing efforts should be devoted to develop the environmentally friendly and low-cost TE materials, such as carbon nanotube (CNT) macrostructures[4–8], conducting polymers[9–14], and CNT/polymer composites[15–27], which attracts considerable attention due to their unique advantages and extensive application prospects. Markedly, the TE power factor (PF) of flexible p-type TE material (the PANI/CNT/graphene multilayer film) has been improved to $\sim 2,000\,\mu W\,m^{-1}\,K^{-2}$ at room temperature[16], which could compete with the state-of-the-art inorganic TE materials[28]. However, high-performance and air-stable flexible n-type TE materials are still deficient, which impedes the performance optimization of TE modules. Further, high-performance flexible n-type materials are also urgently desired in organic solar cells[29], field-effect transistors[30–33] and so on.

$PF = S^2\sigma$ is directly related to the usable power attainable from the TE materials[34], where $S$ and $\sigma$ are Seebeck coefficient and electrical conductivity. n-type organic TE materials is demonstrated for the first time by Sun et al.[12], where metal coordination n-type polymer poly[$K_x$(Ni-ett)] exhibits promising TE characteristics, the highest PF is $66\,\mu W\,m^{-1}\,K^{-2}$. n-type CNTs can be obtained by metal-encapsulated method[25,35], or functionalization with electron-rich polymers[27,33,36–38], small molecules[26] and conjugated polyelectrolytes[24]. In previous researches, all the n-type CNT-based films are fabricated from the dispersed CNT solutions, in which the contacts of intertube junctions are weak and blocked by highly insulating surfactant residues or n-type dopants (as shown in Fig. 1b), resulting in unsatisfactory electrical conductivity and moderate PF (maximum PF is $76\,\mu W\,m^{-1}\,K^{-2}$)[25]. Recently great progress has been made, including the n-type air-stable SWNT sheet fabricated by salt-coordinated method with a good PF of $230\,\mu W\,m^{-1}\,K^{-2}$ (ref. 39), n-type flexible hybrid superlattice of $TiS_2/[(hexylammonium)_x(H_2O)_y(DMSO)_z]$ with a high PF of $450\,\mu W\,m^{-1}\,K^{-2}$ (ref. 40) and the hybrids of CNTs and poly(3,4-ethylenedioxythiophene) treated by tetrakis(dimethylamino)ethylene with an outstanding PF of $1,050\,\mu W\,m^{-1}\,K^{-2}$ (ref. 23). However, the performance of poly(3,4-ethylenedioxythiophene)/CNT films treated by tetrakis(dimethylamino)ethylene is sensitive to ambient humidity and atmosphere, which will hinder their practical application. Therefore, the high-performance (PF $> 1,000\,\mu W\,m^{-1}\,K^{-2}$) and air-stable flexible n-type materials are extremely fascinating.

Here we develop an excellent flexible n-type film based on the single-walled carbon nanotube (SWNT) continuous networks. Excitingly, the n-type films exhibit an ultrahigh TE PF of $\sim 1,500\,\mu W\,m^{-1}\,K^{-2}$ and an outstanding long-term stability in air without encapsulation. Surprisingly, we find that the pristine p-type SWNT films can be rapidly and conveniently switched to n-type films by drop-casting the solution of polyethyleneimine (PEI) in ethanol, which opens an opportunity of industrial production. Inspired by the findings, we design and successfully fabricate a novel-configuration, compact and efficient flexible TE module based on the large-area continuously synthesized CNT films and localized doping technology. The as-prepared compact and flexible TE module with small dimensions of $16\,mm \times 10\,mm \times 0.15\,mm$ can generate a remarkable thermopower of $410\,\mu V\,K^{-1}$. At the hot-side temperature $T_{hot} = 330\,K$ and the $\Delta T = 27.5\,K$ ($\Delta T \leq 30\,K$ is easy to achieve in a natural environment). The TE module exhibits maximum power output of $2.51\,\mu W$, a small internal resistance $R_0 = 12.5\,\Omega$ and a power density of $167\,\mu W\,cm^{-2}$.

## Results

**Preparation of n-type flexible films.** The pristine SWNT films can be directly synthesized by floating catalyst chemical vapour deposition (FCCVD) reported in our previous researches[41,42]. The conductive and continuous network of CNTs was formed in the growth process at high temperature, which results in a distinctive structure and more superior electrical properties. By adjusting the growth parameters, such as the sublimation rates of catalysts and deposition time, we can obtain films with different thicknesses, e.g., 50–500 nm. The synthetic procedure, optical photograph, scanning electron microscope (SEM) and transmission electron microscope (TEM) images of as-grown CNT films are shown in Supplementary Information. Figure 1a schematically shows the fabrication process of n-type SWNT films based on as-grown SWNT continuous networks. First, the pristine SWNT film was transferred onto an ultrathin polyethylene terephthalate (PET; $\sim 2\,\mu m$ in thickness) substrate, which was placed on a glass slide. Subsequently, the solution of branched-PEI (molecular weight 600, 99% from Alfar Aesar) in ethanol was drop-casted into the sample by a micropipette and then the sample was dried at $50\,°C$ in air for 5–10 min. As a result, the fabrication of the n-type SWNT film was accomplished.

**TE properties.** To track this optimum, different doping levels are achieved by varying concentrations of PEI solution drop-casted into several SWNT ribbons ($25\,mm \times 1\,mm$, cut from the same homogeneous as-grown SWNT film with a thickness of $\sim 200\,nm$), ranging from 0.01 to 5 wt.% (while keeping the same solution volume around $30\,\mu l$). The electrical conductivity and Seebeck coefficient of the pristine and doped SWNT ribbons with different PEI concentrations are measured in air at room temperature (1 atm, 25–27 °C, relative humidity (RH): $40 \pm 3\%$ RH). The results are shown in Fig. 2a and the calculated TE PFs are shown in Fig. 2b.

For the pristine SWNT ribbons, owing to the unique conductive reticulate structure and very long and superior Y-type intertube and interbundle junctions (as shown through the schematic diagram in Fig. 1a and the distinct SEM images in Supplementary Figs 3 and 4) formed during the growth process at elevated temperatures[41–43], the measured electrical conductivity ($\sigma \sim 3.02 \times 10^5\,S\,m^{-1}$) is prominent. Therefore, conventional nitric acid treatment can be omitted, which can avoid strong p-type doping effect[44–46], and make SWNT films retain relatively high Seebeck coefficient ($S \sim 78\,\mu V\,K^{-1}$). The positive Seebeck coefficient of SWNT films resulted from oxygen doping in air, which is suggestive of hole-like carriers[47,48]. As a result, the obtained PF is as high as $1,840\,\mu W\,m^{-1}\,K^{-2}$ at room temperature, which is among the highest value ever reported for flexible p-type TE materials[16]. In addition, the SWNT films avoid tedious composite procedure.

For the doped SWNT ribbons, the ethanol is used as the solvent of n-type dopant PEI, which is environmentally friendly and can infiltrate well into the hydrophobic SWNT networks. As a consequence, the amine-rich PEI molecules rapidly and evenly coat on the surface of the nanotubes, which act as highly effective electron donors[33,36]. At low PEI concentrations (0.01 wt.%), a small amount of electrons transfer from the dopant to SWNTs and reduce the hole concentration, resulting in the decrease of the conductivity and the positive Seebeck coefficient. When the PEI concentration is increased to 0.05 wt.%, the Seebeck coefficient of the doped SWNT ribbon turned into a negative value ($-16\,\mu V\,K^{-1}$), indicating the n-type characteristics. The majority carriers of the doped SWNT ribbon is successfully switched from holes to electrons. As the further increase of the

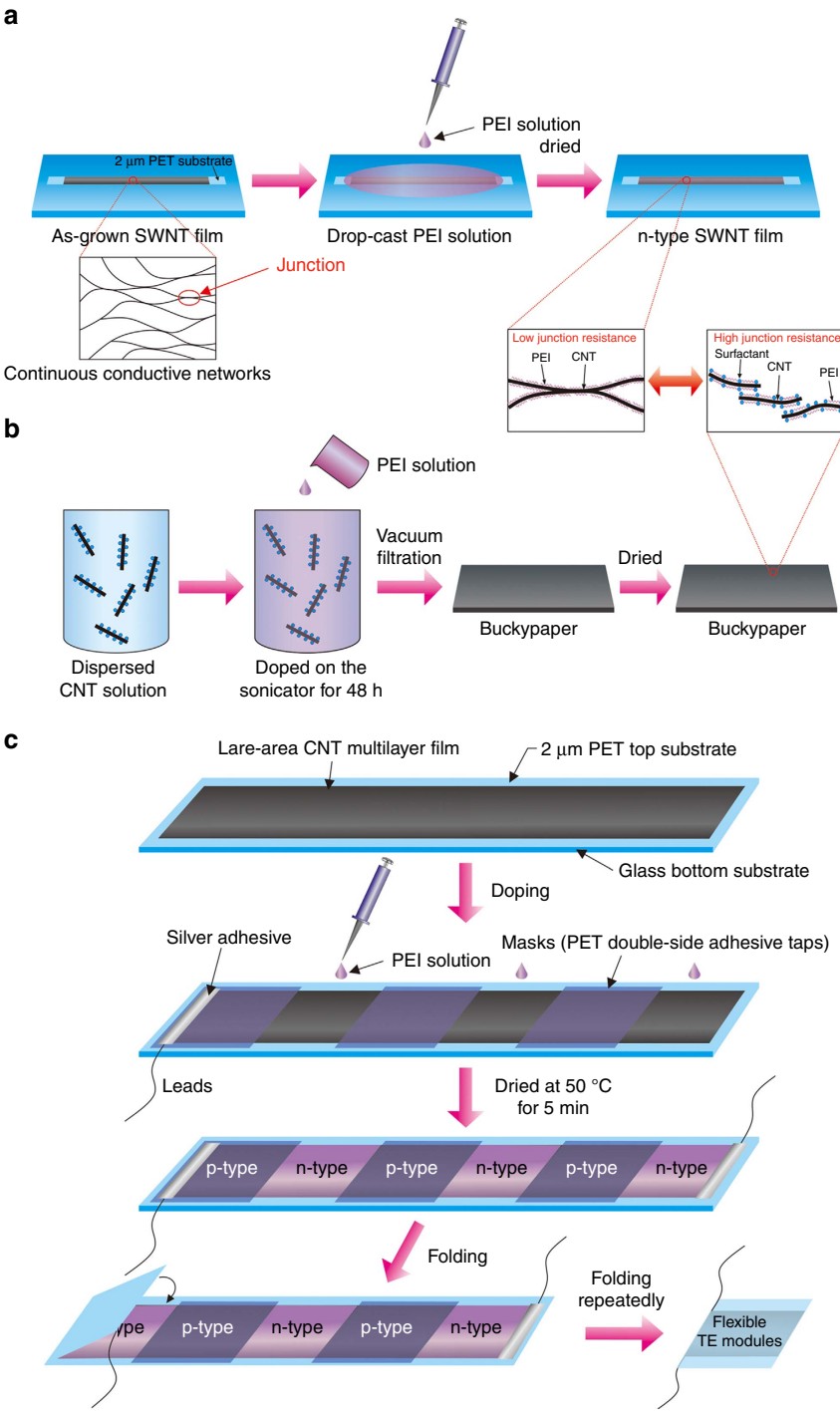

**Figure 1 | Preparation of n-type flexible films and compact-designed TE modules.** Schematics of the fabrication process for (**a**) n-type SWNT films based on as-grown SWNT continuous networks, (**b**) n-type CNT buckypapers based on the dispersed CNT solutions for comparison[36] and (**c**) a novel configuration, compact and efficient flexible TE module based on the large-area continuously synthesized CNT films and localized doping technology.

PEI concentration from 0.1 to 5 wt.%, the $\sigma$ started to increase progressively resulting from more electron injection. On the other hand, the negative Seebeck coefficient increases rapidly at first and reaches a maximum ($-69 \mu V K^{-1}$) at the concentrations of 0.5 wt.%, and then slightly decreases. The dependence of the Seebeck coefficient on the doping level is satisfactorily consistent with the previously reported theoretical prediction[49] and the experimental results[27,37,50].

The mild doping process does not damage the intrinsic reticulate CNT architecture. Therefore, the n-type films inherit the superior electrical conductivity of the pristine SWNT films. The maximal electrical conductivity of the n-type film is $4.68 \times 10^5 S m^{-1}$ at the highest considered doping concentration (5 wt.%). This $\sigma$ is much higher than that of the n-type CNT buckypapers ($2 \times 10^3$–$4 \times 10^4 S m^{-1}$; refs 24–26,36), which are fabricated based on the dispersed CNT solutions. Figure 1b demonstrates the fabrication process of n-type CNT buckypapers for comparison[36], which is convenient for understanding the differences in electrical conductivity between our n-type SWNT films and previously reported n-type CNT buckypapers. It is well

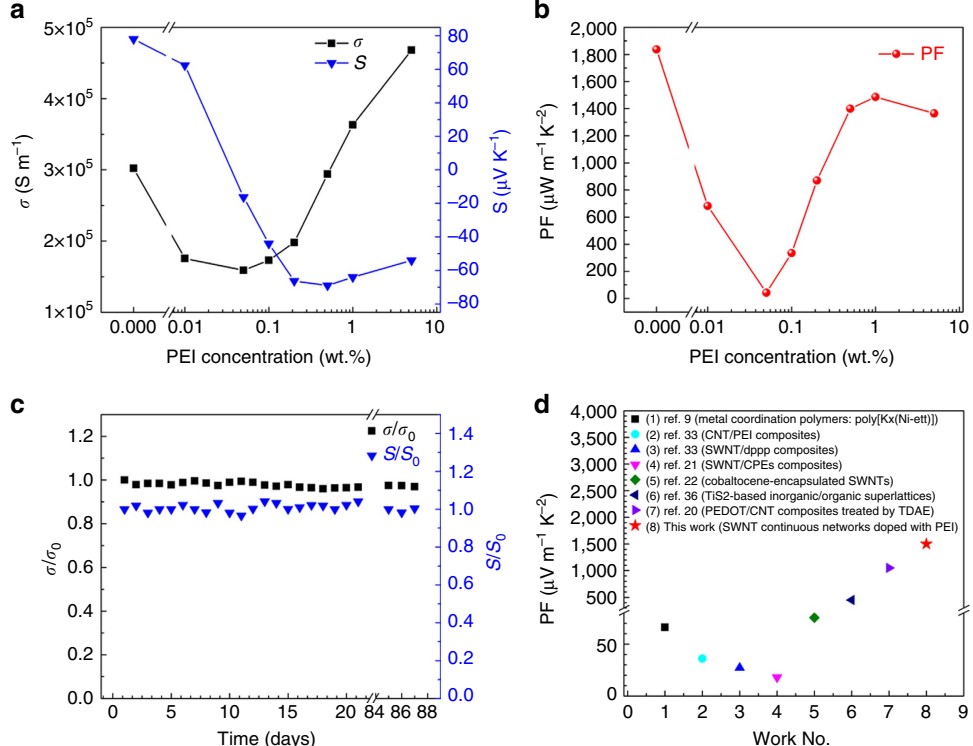

**Figure 2 | TE properties and stability. (a)** The measured electrical conductivities and Seebeck coefficients, and (**b**) the calculated TE power factors of the pristine and doped SWNT films at room temperature. The dopant is a solution of PEI in ethanol with varying concentrations ranging from 0.01 wt.% to 5 wt.%. (**c**) The long-term stability test of the n-type SWNT film doped with 1 wt.% PEI in air without encapsulation. (**d**) The power factors of flexible n-type TE materials in recent researches and our lines of work.

known that the electrical conductivity of the films is dominated by the intertube junctions. For the CNT buckypaper, the contacts of intertube junctions are weak and blocked by insulating surfactant residues and dopant, which hinder carrier transport across the junctions between nanotubes and significantly degrade the electrical conductivity[51]. In addition, ultrasonic dispersing and doping process possibly make CNT shorten or introduce some defects, which also further reduce $\sigma$.

The optimum n-type TE PF ($\sim$1,500 $\mu$W m$^{-1}$ K$^{-2}$) occurs at the PEI concentration of 1 wt.%, which is the highest value ever reported for flexible n-type TE materials[23–25,40]. The PFs of flexible n-type TE materials in recent researches and our works are listed in Fig. 2d. More importantly, the n-type SWNT film exhibits a long-term stability in air as shown in Fig. 2c, which will be discussed later. The ultrahigh PF can be attributed to the prominent electrical conductivity ($3.63 \times 10^5$ S m$^{-1}$) and a reasonably high Seebeck coefficient ($-64 \mu$V K$^{-1}$). Although the $\sigma$ of doped SWNT ribbon is higher with the further increase of the PEI concentration, the PF is slightly decreased because of the relatively low S ($-54 \mu$V K$^{-1}$) based on the expression of PF = $S^2\sigma$.

**Characterization and stability.** SEM images of the pristine and doped SWNT films are shown in Fig. 3a − c. The pristine SWNT network exhibits an excellent continuity and uniformity (Fig. 3a). For the n-type SWNT film doped with 0.5 wt.% PEI, Fig. 3b clearly shows that the surface of the CNTs is coated evenly by PEI; thus, the bundles of CNTs become a little bit thicker than those of the pristine SWNT films. When the PEI concentration was increased to 1 wt.%, more PEI wraps around CNT bundles (Fig. 3c). The atomic force microscopy (AFM) images (Fig. 3d–f) of corresponding samples enable the variations of the

morphology to be observed more intuitively. Figure 3f clearly shows the bundles of CNTs become much thicker at the concentration of 1 wt.%. Almost all of the CNT bundles are coated by PEI polymer chains and the network of CNTs is only faintly visible. The cross-section SEM images of n-type CNT films are provided in Supplementary Fig. 6, which suggest that PEI penetrates inside the CNT films upon doping.

In order to test the stability of the n-type SWNT film doped with 1 wt.% PEI, the as-prepared n-type SWNT film without any encapsulation is fixed on the testing stage for the electrical conductivity and Seebeck coefficient measurements over 3 months. The tests are performed in an air environment (1 atm, 25–27 °C, relative humidity: 40 ± 3% RH) and the results are shown in Fig. 2c. During the test period of 3 months, the variations of electrical conductivity and Seebeck coefficient are less than 5%, which reveals the n-type SWNT films doped with 1 wt.% PEI possess a long-term and outstanding stability in air. It could be understood by speculating that the uniform PEI coating layer acts as a protective coating, which impedes the oxygen doping of the nanotubes in air and preserves the n-type characteristics. N-type CNT buckypapers are fabricated from the dispersed CNT aqueous solutions and PEI, in which high concentrations of surfactant (sodium dodecyl benzene sulfonate) can result in a better-dispersed sample and increase the number of tubes that are evenly coated with PEI. When PEI is coated on most of CNT surfaces, oxygen doping is deterred[27]. Therefore, PEI coating is considered as playing a significant role in maintaining the n-type characteristics. However, previously reported n-type CNT buckypapers doped with PEI without lamination protection exhibits gradual degradation of n-type characteristics in air[36]. To improve the electrical conductivity of as-prepared n-type CNT buckypapers, the doped CNT solution is vacuum-filtered onto a PTFE membrane with additional water to

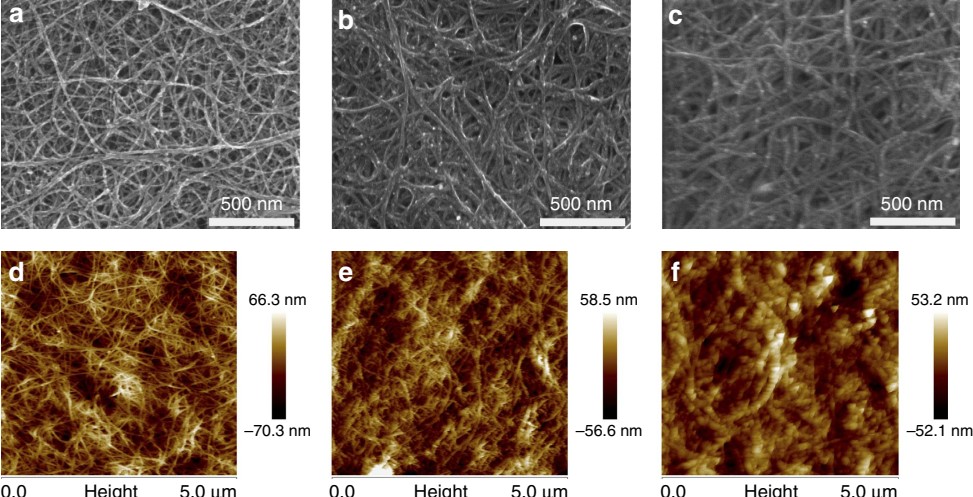

**Figure 3 | Morphology characterization.** (**a–c**) The SEM images, and (**d–f**) the AFM images of the pristine SWNT film (**a**,**d**), the SWNT film doped with 0.5 wt.% PEI (**b**,**e**) and the SWNT film doped with 1 wt.% PEI (**c**,**f**).

remove the excess PEI between CNTs, which probably removes a few of PEI coated on CNTs. Thus, the degradation of n-type characteristics could be attributed to exposure of CNTs during the fabrication process. On the contrary, in terms of our n-type SWNT films, PEI evenly coat on the surface of the whole CNT networks without subsequent rinse procedure.

To further verify charge transfer and doping effect resulting from the PEI doping, the absorption spectra of the pristine and doped SWNT films are measured. Figure 4a shows that the pristine SWNT film has obvious absorption peaks $S_{11}$, $S_{22}$ and $M_{11}$, which are attributed to the electron transitions between van Hove singularities in semiconducting SWNTs and metallic SWNTs[52]. When the SWNT film is doped with PEI, the intensities of semiconducting optical transitions ($S_{11}$ and $S_{22}$) are significantly quenched, which distinctly indicates a strong doping-induced Fermi level shift, which is in agreement with the previous reports[38]. When the PEI concentration increases from 0.05 to 1 wt.%, the attenuations of $S_{11}$ and $S_{22}$ transition become more notable. Absorbance quenching can result from state-filling by holes or electrons[44,53]. Furthermore, combined with the negative Seebeck coefficient of the doped SWNT films, it is demonstrated that the PEI can act as highly effective n-type dopant for CNTs. For a clearer comparison, the absorption spectra removed absorbing background of $\pi$ electron plasmon are shown in Supplementary Fig. 7.

A comparison of normalized Raman spectra excited with a 514 nm laser for the SWNT film before and after 1 wt.% PEI doping are shown in Fig. 4b (the normalized Raman spectra excited with a 633 nm laser are shown in Supplementary Fig. 8). The Raman spectrum of pristine SWNT film shows a tiny intensity of D band (at $\sim$1,350 cm$^{-1}$) compared to that of G band (at $\sim$1,590 cm$^{-1}$), which indicates that the as-grown SWNT film exhibits superior crystal structures and negligible defects[54]. After 1 wt.% PEI doping, intensity of D band remains virtually unchanged, which demonstrates that no structural defect is introduced upon the drop-casting doping process. Contrarily, for the CNT solution-based doping process in previous report[36], a Seebeck coefficient of $-57\,\mu V\,K^{-1}$ is achieved for the n-type CNT buckypaper after 48 h PEI doping in the sonicator and a markedly enhanced D/G intensity ratio occurs after doping, which resulted in a relatively low electrical conductivity of n-type CNT buckypaper. On the other hand, the radial breathing mode, with a frequency lower than 500 cm$^{-1}$, is a bond-stretching out-of-plane phonon mode for which all the carbon atoms move

coherently in the radial direction. The intensity reduction of the radial breathing mode induced by doping indicates the presence of PEI because the PEI attached on CNT surfaces hinder carbon atoms oscillation in the radial direction.

**Preparation of compact-designed TE modules.** The conventional TE module possesses a $\pi$-shaped configuration, where p-type and n-type TE materials are electrically connected in series by metal interconnects. In addition, dozens of nanometre gold or silver top electrodes are usually deposited on each TE legs for decreasing the contact resistance. Flexibility of TE module is crucial to cover closely to the surface of heat sources and take full advantage of ubiquitous waste heat, especially for the heat sources having irregular surfaces. In recent researches, various TE modules based on the CNT-based or organic TE materials are fabricated[9,12,24,25,34,36]. However, the performance of the TE modules is not high enough because the PFs of the constituent TE materials are relatively low. Further, typical $\pi$-shaped configuration that makes the modules have non-compact structures (large intervals exist between each TE legs), which will limit the utilization of heat source.

Inspired by our research results described above, we fabricate a novel-configuration, compact and efficient flexible TE module based on the large-area continuously synthesized CNT films and localized doping technology. Figure 1c shows the fabrication process of the flexible and continuous TE modules. To decrease the internal resistivity of obtained TE module, we prepare large-area thick CNT films having low sheet resistance $\sim$1 $\Omega$ by automatically superposed multilayer continuously produced CNT films and densified with ethanol (the single layer thickness of continuously produced CNT films is $\sim$50 nm, the TE properties of the thick CNT films are shown in Supplementary Tables 1 and 2). First, the large-area CNT films (Fig. 5a) with a thickness of $\sim$3 μm is cut into a 96 mm × 10 mm stripe and placed on an ultrathin PET (2 μm in thickness) substrate, on which thin PET double-side adhesive taps are alternately adhered as masks. Next, the solution of 1 wt.% PEI in ethanol is drop-casted into the unshaded areas and then the stripe is dried at 50 °C in air for 5 min. Therefore, the unshaded areas in CNT films are converted into n-type and the areas covered by PET double-side adhesive taps remain p-type. Subsequently, metal leads are pasted onto the two edges of the stripe by high-purity silver paste for performance test. As a result, three pairs of continuous p–n couples can be

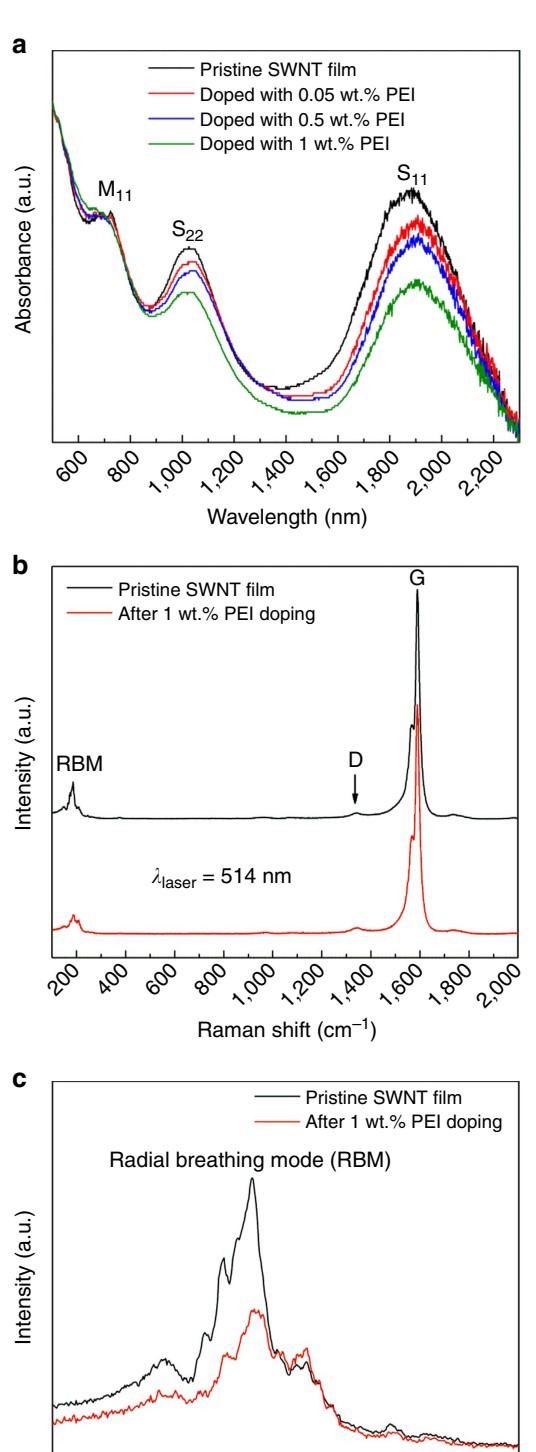

flexibility of the TE module (the whole thickness is ~150 μm). Compared with the conventional π-type TE module, our compact module obviates the need for gold or silver top electrodes deposited on each TE legs and metal interconnects between n-type and p-type legs, and avoids the influence of the contact resistance as well as is easily large-scale produced.

**Performance and demonstration of TE modules.** Owing to the excellent TE properties of the p-type and n-type films based on the as-grown CNT continuous network, the as-prepared TE module exhibits an outstanding performance. When a temperature difference ($\Delta T = T_{hot} - T_{cold}$) is applied between the two ends of the module in the in-plane direction, a noteworthy Seebeck voltage is immediately created. The output voltages in different steady-temperature difference are measured, shown in Fig. 5e. A remarkable thermopower of 410 μV K$^{-1}$ of the module with small dimensions of 16 mm × 10 mm × 0.15 mm is obtained from the voltage–$\Delta T$ curve, which is in reasonable agreement with the theoretically calculated values of ~420 μV K$^{-1}$ by considering that the positive $S$ is ~75 μV K$^{-1}$ for the p-type leg and the negative $S$ is ~−65 μV K$^{-1}$ for the n-type leg. Consequently, the novel-configuration modules can well integrate the excellent TE performance of each TE legs. Figure 5f shows the open-circuit voltage $V_{oc} = 11.3$ mV and short-circuit current $I_{sc} = 0.9$ mA as well as the practical power output at the hot-side temperature $T_{hot} = 330$ K and the steady-state temperature difference $\Delta T = 27.5$ K ($\Delta T \leq 30$ K is easy to achieve in a natural environment). The TE module exhibits maximum power output of 2.51 μW and a small internal resistance $R_0 = 12.5 \, \Omega$. The power density (estimated by dividing the measured $P_{max}$ values by the total cross-sectional area of the module) is 167 μW cm$^{-2}$.

The module performance is superior to previously reported results at the similar temperature gradients, such as a TE module made of three p–n couples based on the p- and n-type CNT buckypapers, which generates a voltage of 6 mV at the $\Delta T = 22$ K and a corresponding power output of 25 nW (ref. 36); a power of 0.3 nW at the $\Delta T = 35$ K by the eight-leg modules fabricated from conjugated polyelectrolytes/SWNT composites[24]; a power density of 0.27 μW cm$^{-2}$ for an all-organic TE module consists of 54 legs[9] and 2.8 μW cm$^{-2}$ for an all-polymer TE module with 35 legs based on poly[Na$_x$(Ni-ett)] and poly[Cu$_x$(Cu-ett)] on AlN substrate at the $\Delta T = 30$ K (ref. 12). Although the five-leg TE module made of p-type n-PETT/CNT/PVC hybrid films generated a high power of 3.88 μW at the $\Delta T = 100$ K (ref. 15), a power of ~0.5 μW at the $\Delta T = 30$ K can be extrapolated for comparison by considering the power out increased linearly with the square of $\Delta T$ (Supplementary Fig. 9).

Figure 6 shows simple demonstrations of using the as-prepared TE module composed of three pairs of continuous p–n couples to generate voltage from the heat resource found easily in daily life (see Supplementary Movies 1 and 2). The demonstration (Fig. 6a–c) illustrates a potential of harvesting biothermal power. When fingers pinch the one end of the module at the room temperature $T_0 \sim 27\,^{\circ}$C, a large Seebeck voltage difference of ~3 mV is rapidly created (Fig. 6b). After removing the fingers, the voltage difference gradually vanishes (Fig. 6c). Flexibility enables the one end of the TE module to be adhered closely to the curved surface of a beaker (Fig. 6d). A voltage difference of ~4.5 mV is rapidly created when the water of 40 °C are pour into the beaker till the liquid level reached the lower edge of the module (Fig. 6f), which indicates common waste heat in the ambient environment can be recovering to electric energy by using our flexible TE module. The temperature of the upper edge of the beaker is higher than the room temperature because of heat

**Figure 4 | Spectroscopic characterization.** (**a**) The normalized absorption spectra of the pristine and doped SWNT films. (**b**) The normalized Raman spectra excited with a 514 nm laser for the SWNT film before (black trace) and after (red trace) 1 wt.% PEI doping.

instantly established in the CNT stripe (Fig. 5b). Finally, a novel TE module (Fig. 5c) is obtained by folding repeatedly the locally functionalized CNT stripe (the operating step is shown in Fig. 1c, the stripe is folded up along the edge lines of PET double-side adhesive taps, which corresponded to the boundary between n-type and p-type region). Figure 5d shows the excellent

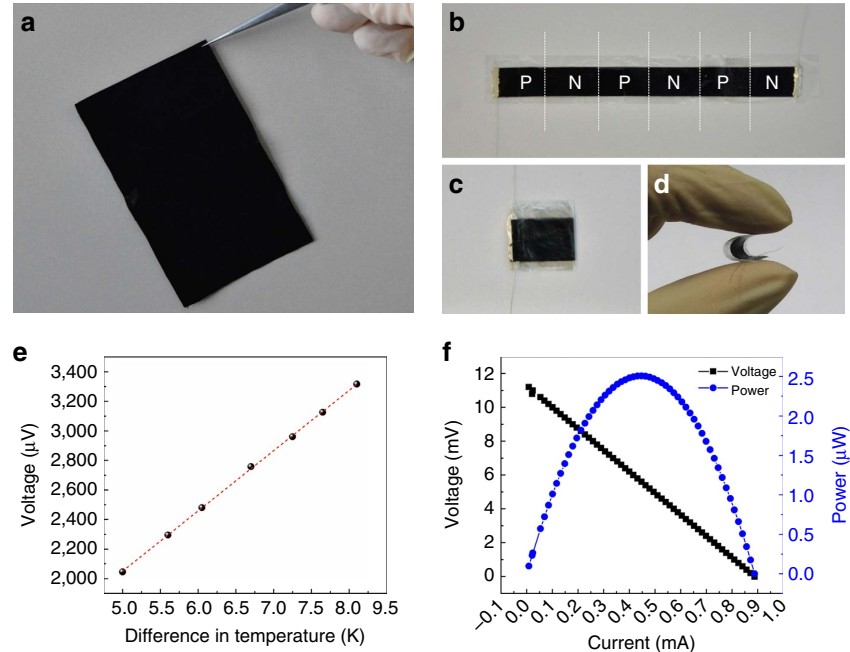

**Figure 5 | Photographs and performance of compact-designed TE modules.** The optical photograph of (**a**) large-area thick CNT films prepared by superposing multilayer continuously produced CNT films and densified by ethanol, (**b**) a CNT stripe composed of three pairs of continuous p–n couples, (**c**) the as-prepared flexible and compact TE module with dimensions of 16 mm × 10 mm × 0.15 mm and (**d**) the flexible display of the TE module. (**e**) The generated voltage in different steady-temperature difference between two ends of the module. (**f**) The voltage–current curve and power–current curve of the module at the hot-side temperature of 330 K and temperature difference of 27.5 K.

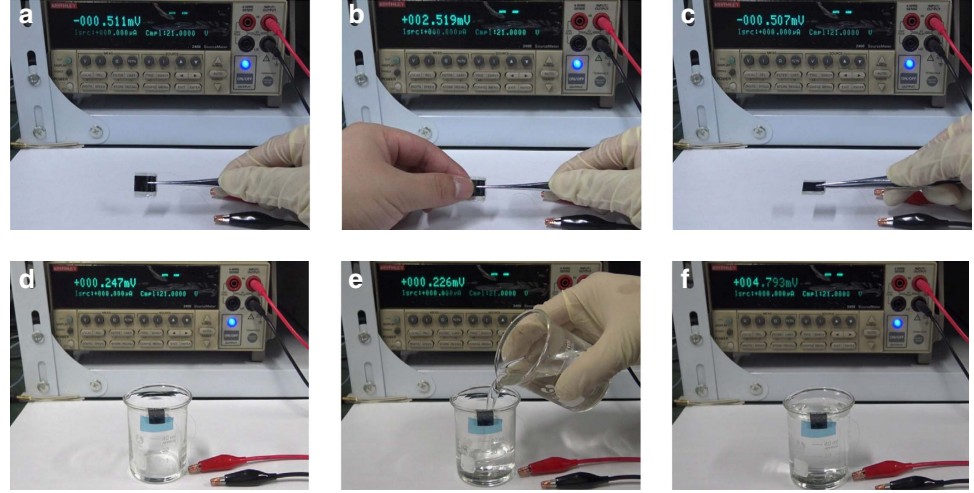

**Figure 6 | Simple demonstrations of the as-prepared TE modules.** (**a–c**) A large voltage difference of ∼3 mV was rapidly created when fingers pinch the one end of the module. (**d–f**) A large voltage difference of ∼4.5 mV was rapidly created when the water of 40 °C were poured into the beaker till the liquid level reached the lower edge of the module. The room temperature is ∼27 °C.

conduction and the actual temperature difference between two ends of the module is ∼10 °C.

## Discussion

This work not only provides an initial demo, but also reveals an extensive prospect. Considering that commercial TE modules consist of hundreds of p–n couples, hence the generated power of our module can easily reach a milliwatt level at a larger temperature gradient by increasing the number of p–n couples in individual module or connecting several modules in series, which can drive small devices, or be capable of cooling. Further,

the as-prepared TE module is highly scalable. For example, a lower internal resistance of the module can be achieved by using thicker CNT multilayer films; thinner masks or deposited insulating layer enable the power density to be enhanced; and enlarge the Seebeck coefficient of pristine CNT films resulting in a higher PF through adjusting the growth conditions.

In summary, the as-grown CNT continuous networks have prominent electrical conductivities and relatively high positive Seebeck coefficients. On the basis of the outstanding host materials, the excellent flexible n-type films are rapidly and conveniently fabricated by drop-casting the solution of PEI in ethanol. The optimum n-type TE performance is obtained by

varying concentrations of PEI solution. The n-type films doped with 1 wt.% PEI exhibit an ultrahigh PF of $1,500\,\mu W\,m^{-1}\,K^{-2}$ and a long-term stability in air (the variations of electrical conductivity and Seebeck coefficient are less than 5% during the test period of 3 months). In addition, we also develop a novel configuration and compact flexible TE module based on the continuously synthesized CNT films and localized doping technology. The performance of our module is markedly superior to that of previously reported flexible TE module. Moreover, we also provide a perspective to further improve our module performance. Given this, the excellent light-weight and flexible p- and n-type TE films and novel-configuration modules based on CNT continuous networks exhibit enormous application potentials in portable and flexible TE conversion as well as novel sensing applications. It is worthy to mention that the research results shown here open opportunities for the upcoming industrial production. For example, combining with a direct and high-yield synthesis technique of continuous CNT films, we can achieve the large-scale and continuous preparation of flexible TE modules by introducing the precisely spraying and automatically folding procedures during the roll-to-roll collecting process of CNT films.

## Methods

**The synthetic procedure of CNT films.** The large-area and freestanding CNT films with superior structure and electrical properties were prepared via a FCCVD method. Experimental details can be found in our previous studies[42,43]. Typically, methane ($\sim 10$ s.c.c.m.) was used as carbon source and argon gas ($\sim 1$ slm) that served as carrier gas. The deposition temperature was $1,000-1,100\,°C$. A mixture of sulfur and ferrocene was used as catalyst. By adjusting the growth conditions, such as the deposition time and sublimation temperature of catalyst, SWNT films with desired thicknesses can be obtained. Recently, our research group has realized continuous and large-scale production of large-area CNT films by a further developed FCCVD technique. The film yield can be as high as hundreds of metres per hour; therefore, we can conveniently obtain thick CNT films with excellent properties by automatically superposed multilayer of CNT films. The preparation mechanism and details will be reported in our next publications.

**Characterization.** The absorption spectra were measured by a ultraviolet–visible–near infrared (NIR) spectrophotometer (UV-3600, Shimadzu). Microscopic morphology was characterized using a SEM (Hitachi S-4800 and S-5200) and atomic force microscope (Bruker MultiMode-8 ScanAsyst).The TEM images were obtained using JEM-2010 instruments. The Raman spectra were recorded using LabRAM HR800 (HORIBA Jobin Yvon Inc.).

**Measurements of TE properties and performance.** The electrical conductivity and Seebeck coefficient in this study were measured in air at room temperature (1 atm, $25-27\,°C$, relative humidity: $40 \pm 3\%$ RH) by a homemade testing stage using the four-electrode method and steady-state method. The in-plane thermal conductivity measurement of CNT films was performed through a developed self-heating method using the homemade measuring apparatus, whose details and results were shown in Supplementary Note 1, Supplementary Fig. 5 and Supplementary Table 3. More details of the measuring apparatus and method can be also found in our recent study[55]. The resistance and voltage were measured using a Keithley 2400 multimetre, and the whole system was controlled by a computer through LabVIEW programmes. The TE performance (including the voltage–current curve and power–current curve) of the as-prepared flexible module fixed on the testing stage at the various steady-state temperature gradient were tested by a Keithley 4200-SCS.

**Data availability.** The data that support the findings of this study are available from the corresponding author upon reasonable request.

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

## Acknowledgements

This work was supported by the National Basic Research Program of China (Grant No. 2012CB932302), the National Natural Science Foundation of China (11634014, 51172271, 51372269 and 51472264), the 'Strategic Priority Research Program' of the Chinese Academy of Sciences (XDA09040202). Dr H.P. Liu thanks support by the Recruitment Program of Global Youth Experts and the '100 talents project' of CAS.

## Author contributions

Wenbin.Z. and Q.F. conceived the idea, discussed and analysed the data. Wenbin.Z. designed and performed the majority of experiments and drafted the manuscript. Q.F. helped the AFM characterizations and performance test of TE modules using Keithley 4200-SCS. Q.Z. and K.L. performed continuous direct production of CNT films by developed CVD method and obtained the large-area thick CNT films. L.C. performed the TEM characterizations. X.G. assisted in synthesizing SWNT films by FCCVD method. F.Y. assisted in the SEM characterizations. N.Z. assisted in the RAMAN spectra measurements. H.L. assisted in measuring absorption spectra. Y.W. assisted in constructing the apparatus of TE property measurements. Weiya.Z. and S.X. critically reviewed and revised the final manuscript as well as proposed valuable advice on project reinforcement. All authors commented on the manuscript and agreed with the results and conclusions.

## Additional information

**Competing interests:** The authors declare no competing financial interests.

