## [Peer Review File · Nature Communications]

Reviewers' Comments:

Reviewer #1 (Remarks to the Author)

The authors describe the fabrication of an air-stable n-type single-walled carbon nanotube (SWNT) sheet by doping with polyethylene imine (PEI). They found that the dropping the SWNT sheet in a PEI solution leads to air-stability, which marked sharp contrast to the previous reports using the dipping method to a PEI solution. The experiments were properly carried out and the conclusions are convincing. However, such n-type air-stability of the SWNT sheet was recently reported by Nonoguchi et al. (*Advanced Functional Materials*, 2016, 26, 3021–3028) although they miss to cite the paper in the manuscript.

Since this point is the main highlight of this study, the impact of the manuscript is not high and publication of this manuscript in *Nature Communication* is not suitable.

Instead, I recommend the authors to submit to a more specialized journal. For the next submission to another journal, the authors are suggested to optimize the dopant quantity when the thickness of the sheet was changed since the amount of the dopant can be controlled in the drop casting method.

Reviewer #2 (Remarks to the Author)

The authors have reported a high-performance and flexible thermoelectric module based on an SWCNT network architecture. To achieve both p-type and n-type CNT conductor, the authors develop a one-step doping method by dropping PEI on as-prepared CNTs. The power factor of n-type CNT can reach as-high-as 1500 $\mu\text{W}/\text{m}^2\text{K}^2$. The manuscript provided a novel thermoelectric module based on lightweight and stable CNT composite and fabricated by a simple folding approach. This may raise wide interests in those researchers who are interested in waste energy conversion. Technically, when thermoelectric conversion was discussed, the manuscript usually needs to provide the performance that covers a rather broad range of temperature, to figure out the optimum working temperature, and the stability under various work conditions, instead of just room temperature performance. Authors also need to give the figure of merit. It is fully understandable that it is quite challenge to determine the accurate thermal conductivity of CNT film in this work. It is very important to evaluate the module based on the efficiency, not just the power output. At this end, the authors should consider revising the manuscript and then resubmit again. Before doing the resubmission, the authors should also make some corrections.

1. There should be a better explanation on the relationship between PEI wt% and the highest power factor. The authors have given a reference (#48) to explain this, but it just provided irrelevant information. In such article, another dopant, BV instead of PEI was used to achieve n-type CNT.
2. In the demonstration of modules (Figure 6), the output is 4.5 mV for a temperature raise of 13 (40-27) degrees. This is obviously much less than what was plotted in Fig. 5e. Should this be attributed to the temperature variance in the sample or in the environment, or the sample inconsistency? Have the authors checked the uniformity of samples?
3. The AFM images actually didn't provide any more valuable information than SEM.
4. The module was fabricated by multiple times of mechanical folding. The authors should explain how to fold the CNT film exactly at the boundary between n-type and p-type region. This is critical to have repeatable experiments.

Responses to the Reviewers' Comments:

Reviewer #1: *The authors describe the fabrication of an air-stable n-type single-walled carbon nanotube (SWNT) sheet by doping with polyethylene imine (PEI). They found that the dropping the SWNT sheet in a PEI solution leads to air-stability, which marked sharp contrast to the previous reports using the dipping method to a PEI solution. The experiments were properly carried out and the conclusions are convincing. However, such n-type air-stability of the SWNT sheet was recently reported by Nonoguchi et al. (Advanced Functional Materials, 2016, 26, 3021–3028) although they miss to cite the paper in the manuscript.*

Since this point is the main highlight of this study, the impact of the manuscript is not high and publication of this manuscript in Nature Communication is not suitable.

Instead, I recommend the authors to submit to a more specialized journal. For the next submission to another journal, the authors are suggested to optimize the dopant quantity when the thickness of the sheet was changed since the amount of the dopant can be controlled in the drop casting method.

Reply : Thank you for reviewing our manuscript and providing valuable comments as well as the latest related reference. We feel sorry that we did not notice this paper. It was published online in April 25, 2016. Our manuscript was drafted in late December of last year and finished in March 2016, afterwards we were so busy making revisions and preparing the submission that we did not pay much attention to the latest advancement. The reference mentioned by you is a nice work and provides interested results and we have cited it as Ref. 39 in the revised manuscript (Page 4).

We apologize if that we did not interpret the highlights and innovations of our work clearly in manuscript. Although such n-type air-stability of the single-walled carbon nanotube (SWNT) sheet was recently reported by Nonoguchi *et al.* (*Adv. Funct. Mater.* **2016**, 26, 3021, hereinafter referred to as *AFM3021*), we think the impact of our novel and comprehensive research on flexible thermoelectric (TE) modules as well as excellent n-type TE materials based on the SWNT interconnected networks is still high and suitable for publishing in Nature Communications based on the reasons below.

First of all, the reports about n-type air-stable CNTs are actually not new. In

previous researches, several doping methods have been demonstrated for the production of air-stable n-type CNTs, such as **Ref.24**, **Ref.25**, **Ref.26** in manuscript. As described in introduction, however, high-performance (Power factor $>1000 \mu\text{W m}^{-1} \text{K}^{-2}$) of air-stable n-type CNTs have never been reported, which impedes the performance optimization of flexible TE modules.

In this work, to achieve a high-performance and flexible TE module, we develop a one-step doping method by drop-casting the solution of PEI in ethanol on as-prepared CNTs. Because of the distinctive micro-structure and superior properties of the as-grown CNT network architecture as well as the efficient PEI doping, the power factor of air-stable n-type CNTs can reach as-high-as $1500 \mu\text{W m}^{-1} \text{K}^{-2}$. The dopant and method used in our manuscript are different from those of **AFM3021**, additionally the obtained in-plane power factor ($1500 \mu\text{W m}^{-1} \text{K}^{-2}$) is markedly higher than that of **AFM3021** ($230 \mu\text{W m}^{-1} \text{K}^{-2}$). The fabrication of the high power factor and air-stable n-type CNTs is only one of highlights in our manuscript.

Most important of all, we further designed a novel flexible thermoelectric module by folding an alternately doped SWNT network film, which is a significant highlight of this manuscript. Because compared with the conventional π -type flexible TE module in previous reports (see **Ref.9**, **Ref.12**, **Ref.24**, **Ref.25** in manuscript, *etc.*), this is an unprecedented design, being compact and without additional metallic interconnects, besides avoiding the influence of the contact resistance as well as being easily large-scale produced. Meanwhile, our compact-designed modules well integrate the superior TE properties of p-type and n-type films based on a reticulate CNT architecture, which results in an attractive high performance compared with previously reported flexible TE modules.

Thus, we believe this manuscript is of great interest not only for scientists in the fields of flexible electronics, organic electronics, novel thermoelectric materials and modules, as well as fundamental researches of carbon nanotubes, but also for engineers in the fields of thermoelectric conversion, sensing applications, and so forth.

Point 1: *Instead, I recommend the authors to submit to a more specialized journal. For the next submission to another journal, the authors are suggested to optimize the dopant quantity when the thickness of the sheet was changed since the amount of the dopant can be controlled in the drop casting method.*

Reply : Thanks for the reviewer's advice. When the thickness of the CNT sheet was

changed, we also optimized the dopant quantity. For example, to decrease the internal resistivity of obtained TE module, the large-area and relatively thick CNT film was used for fabricating the compact-designed module, which has a thickness of $\sim 3 \mu\text{m}$. Before alternately doping the CNT film, we performed experiments to track this optimum dopant quantity for the thick film, where varying concentrations of PEI solution were drop-casted into several CNT ribbons ($25 \text{ mm} \times 1 \text{ mm}$, cut from the same homogeneous CNT film with a thickness of $\sim 3 \mu\text{m}$). The corresponding experiment results are added in Supplementary Table 2.

Reviewer #2: *The authors have reported a high-performance and flexible thermoelectric module based on an SWCNT network architecture. To achieve both p-type and n-type CNT conductor, the authors develop a one-step doping method by dropping PEI on as-prepared CNTs. The power factor of n-type CNT can reach as-high-as 1500 $\mu\text{W}/\text{m}/\text{K}^2$. The manuscript provided a novel thermoelectric module based on lightweight and stable CNT composite and fabricated by a simple folding approach. This may raise wide interests in those researchers who are interested in waste energy conversion. Technically, when thermoelectric conversion was discussed, the manuscript usually needs to provide the performance that covers a rather broad range of temperature, to figure out the optimum working temperature, and the stability under various work conditions, instead of just room temperature performance. Authors also need to give the figure of merit. It is fully understandable that it is quite challenge to determine the accurate thermal conductivity of CNT film in this work. It is very important to evaluate the module based on the efficiency, not just the power output. At this end, the authors should consider revising the manuscript and then resubmit again. Before doing the resubmission, the authors should also make some corrections.*

Reply: Thank you for your valuable comments. We fully agreed with your opinion. This is also our main considerations. The figure of merit of thermoelectric (TE) materials is very important to evaluate the module. Just as what you said, at present it is quite challenge to determine the accurate thermal conductivity of CNT film in our work, which requires specific and advanced measuring techniques. It is for this reason that previous researchers usually estimated the ZT by measuring out-of-plane thermal conductivity of CNT films, while the power factor was measured in the in-plane direction. The obtained ZT is actually inaccurate because of the anisotropy in the film. Here, we tried to measure the in-plane thermal conductivity of CNT films using the home-made measuring apparatus through a developed self-heating method. The details and results of thermal conductivity measurement are added in Supplementary Information (Supplementary Note 3, Supplementary Figure 6 and Supplementary Table 3). The estimated in-plane ZT ~ 0.02 of the as-prepared n-type CNT film in our work can be calculated based on the measurement result of the in-plane thermal conductivity ($18 \text{ W m}^{-1} \text{ K}^{-1}$), which is higher than the recently reported in-plane ZT ~ 0.002 of n-type air-stable CNT sheet prepared by a salt-coordinated method (*Adv. Funct. Mater.* **2016**, 26, 3021).

Thank you for the comment on the optimum working temperature. Currently inorganic semiconductors are the most commonly studied TE materials, such as lead and bismuth telluride, which possess high-temperature resistance. Thus researchers usually provide the TE performance cover a rather broad range of temperature, to figure out the optimum working temperature. Generally, polymers used in organic thermoelectric researches are difficult to suffer from the temperature beyond 150 °C. In most of previous researches, the performances of organic thermoelectric devices were measured near room temperature (such as **Ref.9**, **Ref.24**, **Ref.25**, **Ref.36** in manuscript). Therefore, the as-prepared flexible thermoelectric module based on the CNT and polymer composites in our work is considered to be suitable for thermoelectric conversion near room temperature.

Point 1: *There should be a better explanation on the relationship between PEI wt% and the highest power factor. The authors have given a reference (#48) to explain this, but it just provided irrelevant information. In such article, another dopant, BV instead of PEI was used to achieve n-type CNT.*

Reply: Thanks for your valuable comments. We apologize that we did not interpret the relationship between PEI wt.% and the highest power factor clearly in manuscript. It is the main reason why we cited **Ref.48** that is to illustrate the dependence of the Seebeck coefficient on the doping level. Although **Ref.48** used another n-type organic dopant BV in experiments, however the inset of Figure 6 in **Ref.48** shows the doping-level dependence of the CNT Seebeck coefficient based on theoretical calculation (*Scientific Reports* **2013**, 3, 01335). As the n-type doping is increased, the value of Seebeck coefficient increases rapidly at lower doping concentrations, reaches a maximum at intermediate concentrations, and then slowly decreases, which is consistent with the PEI experimental results in our work. Additionally, other references (**Ref.27**, **Ref.37**) using PEI as n-type dopant also exhibit similar dependence of the Seebeck coefficient on the doping degree. We have added the related references mentioned above in the revised manuscript as Refs. 49 (Page 7).

Point 2: *In the demonstration of modules (Figure 6), the output is 4.5 mV for a temperature raise of 13 (40-27) degrees. This is obviously much less than what was plotted in Fig. 5e. Should this be attributed to the temperature variance in the sample or in the environment, or the sample inconsistency? Have the authors checked the uniformity of samples?*

Reply: Fig. 5e shows the generated voltage in different steady temperature difference between two ends of the module. The steady temperature differences were accurately measured by T-type thermocouples attached two ends of the module. When in the demonstration of modules, a difference of 13 (40-27) °C is the temperature difference between the environment and the water. As shown in Fig. 6f, most of the water has been poured into the beaker, which would result in the temperature of the upper edge of the beaker is higher than the room temperature because of heat conduction. Thus, the actual temperature difference between two ends of the module should be smaller than 13 °C and the output voltage of 4.5 mV is reasonable.

Point 3: *The AFM images actually didn't provide any more valuable information than SEM.*

Reply: We agree with the reviewer's opinion. The AFM images actually didn't provide any more valuable information than SEM. Compared to the AFM images in our manuscript, the SEM images are the enlarged views of surface morphology on a smaller scale. The AFM images not only show the bundles of CNTs become much thicker with the increase of PEI concentration, but also exhibit the change of the surface smoothness.

Point 4: *The module was fabricated by multiple times of mechanical folding. The authors should explain how to fold the CNT film exactly at the boundary between n-type and p-type region. This is critical to have repeatable experiments.*

Reply: Thanks for your advice. In the manuscript, the folding operation is concretely shown in Fig. 1c. As you suggested, the description of how to fold the CNT film at the boundary between n-type and p-type region has been strengthened in the revised manuscript (Page 13).

Reviewers' Comments:

Reviewer #1 (Remarks to the Author)

The authors have modified the manuscript based on the comments of the reviewers. However, this reviewer still considers that the manuscript does not meet the criteria for publication in Nature Communication. Thus this reviewer suggests the authors to submit this manuscript to Scientific Reports.

Reviewer #2 (Remarks to the Author)

The resubmitted revised manuscript, authored by Zhou, W. B. et al is improved in comparison with the original one that has been submitted in September. Currently, the primary concern of the suitability of this paper to be published in Nature Communications is the merit of this work, which has been criticized by referees. In this submission, authors have performed a thorough literature review and concluded this flexible thermoelectric module based on reticulate CNTs actually delivered much higher power factors among all relevant modules. As a result, it is suggested that this work will present considerable technical breakthroughs in the field of flexible waste heat recovery devices. Furthermore, the authors also included the figure of merit of the doped and undoped CNT films. This improvement will be a valuable part to compare with state-of-the-art modules based on other inorganic solids, e.g. alloys. Before the manuscript is published, I will suggest several issues should be further addressed.

1. It is suggested for authors to avoid any claim such as "for the first time" in the manuscript, because PEI-doped CNT film for n-type conductors is not originally new at this stage.
2. It is better to provide any pertinent references for the thermal conductivity measurements in the supporting information, to make this part more accessible for a wider range of readers.
3. In the reply, authors have attributed the lower thermoelectric voltage of CNT modules in the hot water to the heated edge by the water. By claiming this, the authors should also provide corresponding corrections to this systematic error, as they have known the temperature difference is less than 13 C.
4. Larger fonts should be considered to make Figure 2 d more readable.

Responses to the Reviewers' Comments:

We appreciate reviewers for their valuable comments and suggestions. According to the reviewers' suggestions (marked in italics in this response letter), we have taken all of them into careful consideration and made corresponding revisions using the 'track changes' feature in word. Below are the point-to-point responses to the reviewer's comments.

Reviewer #2 (Remarks to the Author):

The resubmitted revised manuscript, authored by Zhou, W. B. et al is improved in comparison with the original one that has been submitted in September. Currently, the primary concern of the suitability of this paper to be published in Nature Communications is the merit of this work, which has been criticized by referees. In this submission, authors have performed a thorough literature review and concluded this flexible thermoelectric module based on reticulate CNTs actually delivered much higher power factors among all relevant modules. As a result, it is suggested that this work will present considerable technical breakthroughs in the field of flexible waste heat recovery devices. Furthermore, the authors also included the figure of merit of the doped and un-doped CNT films. This improvement will be a valuable part to compare with state-of-the-art modules based on other inorganic solids, e.g. alloys. Before the manuscript is published, I will suggest several issues should be further addressed.

1. It is suggested for authors to avoid any claim such as “for the first time” in the manuscript, because PEI-doped CNT film for n-type conductors is not originally new at this stage.

Reply : Thanks for your advice. We have revised the expression in manuscript and delete the phrase “for the first time”.

2. It is better to provide any pertinent references for the thermal conductivity measurements in the supporting information, to make this part more accessible for a wider range or readers.

Reply : We fully agreed with the reviewer's opinion. We have added pertinent references for the thermal conductivity measurements in the Supplementary Information.

3. In the reply, authors have attributed the lower thermoelectric voltage of CNT modules in the hot water to the heated edge by the water. By claiming this, the authors should also provide corresponding corrections to this systematic error, as they have known the temperature difference is less than 13 C.

Reply : Thank you for the valuable comment. We have revised related content in the manuscript (Page 16).

4. Larger fonts should be considered to make Figure 2 d more readable.

Reply : Thanks for the reviewer's advice. We have enlarged the legend in figure 2d to make them more readable.